# The Effect of the COVID-19 Pandemic on the Assessment of Sexual Life—Repeated Cross-Sectional Surveys among Polish Adults in 2017, 2020 and 2021

**DOI:** 10.3390/ijerph19074110

**Published:** 2022-03-30

**Authors:** Zbigniew Izdebski, Jolanta Słowikowska-Hilczer, Joanna Mazur

**Affiliations:** 1Department of Biomedical Aspects of Development and Sexology, Faculty of Education, Warsaw University, 00-561 Warsaw, Poland; 2Department of Humanization in Medicine and Sexology, Collegium Medicum, University of Zielona Gora, 65-729 Zielona Gora, Poland; 3Department of Andrology and Reproductive Endocrinology, Medical University of Lodz, 91-425 Lodz, Poland; jolanta.slowikowska-hilczer@umed.lodz.pl

**Keywords:** cluster analysis, COVID-19 pandemic, Polish population, repeated surveys, sexual activity, sexual difficulties, sexual needs, sociodemographic factors

## Abstract

The study aims to investigate whether assessment of sexual life remained stable during the COVID-19 pandemic. Two surveys were conducted among Polish adults aged 18–70 years in June 2020 (*n* = 2042; perspective of last 2–3 months) and in June 2021 (*n* = 2418; last 12 months). Data from 2017 (*n* = 1980) were used as a reference point. Four questions allowed for defining five sexual life assessment profiles (k-means cluster analysis). Their characteristics were presented using 12 variables and 16 factors that contributed to difficulties in sexual life. The 2020 survey showed a temporary increase in the importance of sexual life and the frequency of sexual intercourse. However, the percentage of respondents representing the most favorable profile decreased significantly over the consecutive survey periods (47.1%, 34.2%, and 32.3%, respectively). Pandemic-induced fatigue and stress as well as the permanent presence of others at home were reported as two main factors negatively affecting the frequency of sexual intercourse during the pandemic. Respondents who assessed their sexual life as poor were more likely to consider illness, depression, and low self-esteem as factors negatively impacting their sexual life in 2021 than a year earlier. The results confirmed that as the pandemic drew on, the assessment of sexual life changed compared to the time around the first lockdown.

## 1. Introduction

An increasing number of COVID-19 cases has been observed in Europe since the beginning of 2020. Lockdown due to the COVID-19 pandemic was enforced by the Polish government in early March. The lockdown was unexpected and beyond the control of individuals. It generated a feeling of uncertainty about the future due to financial losses, insecurity about having adequate stores of food and other supplies, separation from loved ones, and general social distancing. The risk of illness and death, as well as frightening news, and inadequate or conflicting information, were sources of intense stress. Many adults worked from home, often without a separate quiet space. Household members had to spend more time together, often sharing a limited living area. Children participated in remote learning from home, and parents were responsible for supervising their education and leisure time activities. Adolescents and young adults faced increased parental monitoring, reduced privacy, diminished independence, and difficulties in physical interaction with peers [1,2].

Some people who have potentially been exposed to a SARS-CoV-2 virus were in quarantine in order to reduce the risk of infecting others. Although social distancing restricts movement less than a quarantine, in both cases, the same psychological effects occurred, such as anxiety, depression, post-traumatic stress, frustration, confusion, and anger [3,4,5,6]. In the group of young adults (18–23 years of age), females were at more risk for increased loneliness and depression compared to males [7]. The COVID-19 pandemic also affected reproductive health, causing problems with access to contraception, reproductive health services, such as prenatal and postnatal care, childbirth and abortion services, and the management of sexually transmitted infections [8,9,10].

The imposed social distancing might have had an impact not only on psychological distress, but also on a quality of life. The population most at risk for this adverse impact were people from sexual and gender minorities [11,12,13,14]. Moreover, a relatively higher prevalence of psychological, physical, and sexual violence against women was noted [15,16,17]. Observations in Germany and the United States demonstrated that both women and men experienced more anger and aggression, and they predominantly directed their anger at others, rather than themselves [18,19].

Sexual life as a component of sexuality includes, among others, sexual behavior and preferences, partner relationships, gender roles, as well as sexual identity and orientation. Sexual life is influenced by attitudes and beliefs as well as by the current developmental stage. Many research tools are used to assess the sexual life single of people or those living in different types of relationships—monogamous and non-monogamous. When addressing sexual life, attention is drawn not only to the frequency and type of contacts, number and gender of partners, clinically assessed dysfunctions, but also to the subjective assessment of sexual performance, the discrepancy between expectations and their fulfillment, the general level of satisfaction, experienced sensual pleasure, sexual fantasies, and quality of sexual life in the various dimensions indicated above. A significant number of the above components of sexual life were analyzed in relation to the pandemic period [20,21,22,23,24,25,26,27,28]. The contributions came from various countries, including Poland [19,25,29,30,31,32,33,34], sometimes based on multi-country projects [8,23,35].

Because of increased stress during the pandemic, the quality of sexual life and partner relationship also deteriorated [36,37]. Cito and colleagues [29] showed that most quarantined participants reported reduced frequency of sexual intercourse per week, but not autoeroticism. Moreover, men presented lower sexual desire than women. However, higher rates of sexual dysfunction and reduced sexual activity were observed more frequently in women compared to men [24]. In turn, Fuchs and colleagues [38] revealed a decrease in the overall Female Sexual Function Index (FSFI), as well as scores in domains such as desire, arousal, lubrication, orgasm, and satisfaction. Some authors [30,31,39] reported that living in a metropolitan area was associated with a decline in both sexual desire and sexual intercourse frequency during the COVID-19 pandemic. However, it was found that a satisfying sexual life in both males and females had a protective effect against the worsening of mental health and quality of life, which were the expected effects of the pandemic [20,40]. Yuksel and Ozgor [21] showed that sexual desire and frequency of intercourse increased significantly in Turkish females during the COVID-19 pandemic, which can be explained by more time spent at home with a partner. However, the quality of sexual life significantly decreased. Moreover, the pandemic was associated with less desire for pregnancy, decreased female contraception, and increased menstrual disorders.

While interpreting the results of studies into the health impacts of the COVID-19 pandemic, one should consider the time of research and the successive waves of increase and decrease in the frequency of infections. From a long-term perspective, we can expect adaptation to new living conditions and an increased sense of safety after the introduction of mass vaccination [41], but also increasing symptoms of persistent anxiety, depression, and post-traumatic stress [42]. When interpersonal relationships are concerned, differences in the functioning of formal or informal relationships and sexual life can be expected between the first and subsequent months of the pandemic.

Poland is one of the European countries with persistent and extremely unfavorable rates of recorded infections and deaths related to the SARS-CoV2 virus [43]. Several scientific centers have undertaken research into the health impacts of the pandemic in Poland, focusing on mental and physical health, consequences of the infection, less frequent health-enhancing behaviors, and more frequent use of stimulants [44,45]. Changes in attitudes toward social values have been pointed out as a likely result of living in the COVID-19 pandemic. The general slowing down of life is conducive to reflecting on its meaning and changing how we look at many issues. Putting a higher value on issues such as health, freedom, and family may have occurred in the face of their deprivation, and strong relationships with loved ones may enhance resilience and coping strategies [46,47].

Our study aims to assess the change in the evaluation of sexual life in Polish adults, comparing the first months of the pandemic with the period after another year of its duration. Moreover, data from 2017 were used as a reference point. We decided to identify patterns of assessment of sexual life (hereinafter called profiles) and assess the frequency of their occurrence in the Polish population. It can also be hypothesized that the first months of the pandemic were associated with rapid changes, after which the indicators of sexual life may have plateaued or returned to their previous levels. The specific aims of the study were:to identify sexual life assessment profiles, taking into account the complex relationship between overall evaluation (satisfaction and the meaning of sex in one’s life) and sexual needs (level and fulfilment);to explore how the share of each profile has changed over time between 2017 and 2021;to describe those profiles in terms of socio-economic characteristics, sexual activity, and subjective assessment of sexual performance;to identify the most common problems related to sexual activity during the pandemic and their relationship to the sexual life assessment profile.

## 2. Materials and Methods

### 2.1. Study Design and Sample

The present study used the results of three cross-sectional surveys of the health and sexuality of Polish adults carried out in 2017, 2020, and 2021. The 2017 study was the sixth round of a recurring cycle of analyses related to the subject conducted as a pen and paper personal (PAPI) interview [48]. The 2020 and 2021 surveys were ad hoc online studies (CAWI—computer assisted web interview), organized in connection with the COVID-19 pandemic by the same researchers as in 2017 and the same external company, which was responsible for drawing samples and supervising the field study process. Those three studies were approved by the Scientific Research Ethics Committee of Warsaw University (Nr 5/2020 and 6/2020, 9/2021, respectively). Subsequent surveys included 2500, 3000, and 2500 Polish adults aged 18 and older. For further analyses, 6440 respondents were qualified, who met the following criteria: age up to 70 years, sexual initiation, and complete information in the key four questions on evaluation of sexual life. The mean age of the subjects in subsequent rounds was 39.6 ± 13.7, 43.4 ± 14.4, and 44.7 ± 14.2 years. Those surveyed in 2017 were clearly younger, but this also means that the three groups were from close birth cohorts. In order to ensure comparability of the results of the three studies, we also excluded respondents who in one of these questions declared a complete lack of sexual needs. Such an answer implied skipping the next question about the level of needs fulfillment (in 2020 and 2021, but not in 2017). A flowchart describing sample selection is provided in the Appendix A (Figure A1). The final sample included 2042 respondents from 2017, 2418 from 2020, and 1980 from the 2021 wave, representing 82–83% of the original samples.

### 2.2. Survey Measures

Approach to sexual life was assessed using the four questions previously used in Polish research [38]:


*What is the role of sex in your life at present? With five categories of answers, from entirely unimportant to definitely important.*



*How satisfied have you been with your sexual life over the past 2–3 months/12 months? With five categories of answers, from completely dissatisfied to very satisfied.*



*How do you assess the level of your sexual needs over the past 2–3 months/12 months? With six categories of answers, from I have no sexual needs to I have very high sexual needs.*



*Have your sexual needs been fulfilled over the past 2–3 months/12 months? With five categories of answers from definitely not to definitely yes.*


The last three questions differed in time perspective across the survey periods. In 2017 and 2021, they related to experiences over the past 12 months, while in 2020, to experiences over the past 2–3 months. This assumption allowed for a better assessment of the effect of the pandemic.

Difficulties in engaging in sexual intercourse were analyzed with the help of the following question: *Have you experienced any difficulties or disturbances engaging in sexual intercourse over the past 2–3 months?* The respondents marked any of the 16 problems which may have occurred during the pandemic, while only 12 in 2017. Additional difficulties that may have arisen during the pandemic, such as fear of coronavirus infection and the constant presence of children at home, were pointed out.

When describing the studied groups, the following were taken into account:level of education recoded from 12 categories to three: below secondary, general secondary and vocational, higher than secondary;place of residence recoded from seven to four categories of settlement size: rural areas, small towns with up to 100,000 inhabitants, medium-sized towns, large towns with more than 500,000 inhabitants;employment status divided into employed (5 employment categories taken together) and other (including unemployed, studying, retired, housewife, or raising children);family wealth in three groups recoded from five categories, where 1—not enough money to live on or an ability to pay only for the most important needs; 2—enough money for daily expenses but not for major expenses; 3—enough money for major expenses and to put aside savings;attitude towards religion recoded from six categories, with a division into believers and non-believers;relationship status, broken down into singles and people living in formal and informal relationships;sexual orientation recoded into two groups: heterosexuals and others;self-rated sexual performance recoded into four categories: poor, average, good, and very good;date of last sexual intercourse recoded into four categories: this week, this month, this year, earlier or missing data.

The above questions were identically worded across the three studies. In 2017, only the presence of chronic diseases was asked in a different way, using the so-called categorial approach [49].

### 2.3. Data Analysis

Most of the comparisons applied to contingency tables and the significance of differences was tested using the chi-squared test for categorized data. Missing data and response refusal were omitted in layout tables and testing. For tables with more than two rows or columns, values of standardized residuals were analyzed. The distribution of answers to four questions about the evaluation of sexual life was compared, and following that the profiles of sexual life assessment were distinguished, using the k-means method. Each respondent was allocated to one of the “k” groups that was closest to him or her (based on Euclidean distance) using the iterative method. At first, five groups (clusters) were defined on the basis of 2017 data. When cluster membership was examined in 2020 and 2021, the 2017 cluster centroids were adopted as basis for the allocation. In order to decide their optimal number, the results of the post hoc Tukey’s test in the analysis of variance were used. The clusters were approximately arranged in the order from the worst to the best, according to the external criterion of general assessment of sexual life provided in another part of the questionnaire (rating different aspects of life on a 5-point scale from *very bad* to *very good*). Demographic and social characteristics of the clusters were presented on the basis of the 2021 pandemic sample, recognizing that these are the most current data.

Difficulties in engaging in sexual intercourse were arranged in terms of their frequency in the 2021 sample. Their occurrence in individual clusters was also examined.

The expected and observed cluster membership was compared using the kappa coefficient. Theoretical cluster membership was determined by the threshold model (ordinal regression), estimated by PLUM procedure (Polytomous Universal Model). The models also helped to assess whether differences between study periods in profiles distribution persist after adjusting the analyses for other factors.

All reported *p*-values were two-tailed, and values ≤ 0.05 were considered significant. The calculations were made using the IBM SPSS Statistics for Windows software (Version 27.0, IBM Corp., Armonk, NY, USA).

## 3. Results

### 3.1. Characteristics of the Participants

The characteristics of the respondents in 2017, 2020, and 2021 have been presented in Table 1. When comparing all three study periods, no significant differences were found, only according to gender (50.1% females in all combined samples). The 2017 sample differed significantly from the 2020 sample, while fewer differences were found when comparing the 2020 and 2021 waves. Between 2020 and 2021, the percentage of respondents from relatively more affluent families increased and the percentage of respondents identifying themselves as believers decreased. In addition, respondents reported less frequent sexual intercourse and lower ratings of their sexual performance.

A comparison of the 2017 sample and the combined 2020–2021 sample shows that respondents surveyed during the pandemic:were significantly older; the differences were seen in the marginal age groups, with a stable proportion of those aged 30–49 year, (*p* < 0.001);were less educated, although with a stable share of people with higher education, (*p* < 0.001);were less likely to be employed, (*p* < 0.001);more often lived in smaller cities, with a stable share of people from large cities, (*p* = 0.017);were less likely to describe themselves as religious, (*p* < 0.001);more often came from poorer families, with a stable share of people from the most affluent group, (*p* < 0.001);were less likely to live alone and more likely to live in informal relationships, with a stable share of married people, (*p* < 0.001).

Differences were also noted for questions more closely related to sexual life. In comparison with 2017 data, respondents surveyed during the pandemic:were less likely to declare heterosexual orientation, (*p* = 0.012);were found to be less sexually active reporting their last sexual intercourse more than a month ago (*p* < 0.001);rated their sexual performance lower, (*p* < 0.001).

### 3.2. Assessment of Sexual Life

Table 2 presents a comparison of the assessment of sexual life in the 2017, 2020, and 2021 studies, presented as a percentage of marginal answers to four core questions. In 2021, 55.1% respondents meeting the inclusion criteria were of the opinion that sex plays an important or very important role in their lives, 61.1% were very satisfied or rather satisfied with their sexual life, 34.0% had very high or high sexual needs, and 67.2% considered their needs fulfilled. These indicators confirmed lower sexual life satisfaction during the pandemic than in 2017 and lower levels of fulfilment of sexual need. The importance of sexual life and the level of sexual need satisfaction were similar in 2021 and before the pandemic. Three time points allow us to assess the trajectory of change. The role of sex clearly increased in the first few months of the pandemic, then declined again. Satisfaction with sexual life worsened systematically, which was reflected in the higher percentage of people who gave a negative answer. The level of sexual needs proved to be the most stable indicator. Fulfilment of these needs deteriorated systematically when comparing successive time points.

### 3.3. Profiles of Satisfaction with Sexual Life

Based on the four questions concerning satisfaction with sexual life, five profiles were distinguished. The average scores for each of the four questions, five clusters, and three time points have been presented in Appendix A).

Profiles involving positive evaluation are significantly more common in the population under study. Classification appears to be valid, as the division is consistent with the general assessment of sexual life analyzed in another section of the questionnaires by means of a separate question. The percentage of participants who rated their sexual life as very good or rather good in 2021 was 53.0%, increasing from 4.8% to 87.9% in subsequent clusters (Figure 1).

A more detailed characteristic of these profiles is presented in Table 3. In the group considered the least favorable, there are negative answers to all questions, and positive answers sometimes do not even occur. The opposite situation is observed in the fifth—the best group. Each of the middle groups has specific problems. In the second (still relatively worse) group, there is a low level of satisfaction with sexual life and low fulfilment of sexual needs, despite the fact that a significant number of respondents declare a high level of needs and consider sex important in their life. Group three is characterized by low levels of sexual needs and low importance of sex in their life. Group four is characterized by a very high level of satisfaction with sexual life, but low level of sexual needs. Respondents in this group avoided marginal answers to question three, and 82.6% rated their needs as average (category not shown in Table 2 and Table 3).

In a more narrative way, the groups could be labeled as follows:


Profile 1–Low rating on all four aspects, including sex not being an important part of life;Profile 2–Low rating of satisfaction with sexual life and level of fulfilment of needs, however, sex is rather important in life and level of needs rather high;Profile 3–Sexual needs satisfied at a relatively high level, however, sex does not play an important role in life and the level of needs is low, avoidance of marginal ratings of overall satisfaction;Profile 4–High level of satisfaction with sexual life and level of fulfilled needs, relatively high importance of sex in life, however, the level of sexual needs at average level (rare marginal ratings);Profile 5–High rating in all aspects, including sex playing an important role in one’s life.


A comparison of 2017, 2020, and 2021 data indicates a fairly stable structure of belonging to certain profiles, but also some differences (Table 4). The percentage of people classified into the three least favorable profiles remained at a similar level. As the analysis of residuals in the contingency tables shows, the greatest differences concern groups 4 and 5. Both in 2020 and 2021, the percentage belonging to the most favorable profile was lower than before the pandemic in 2017. At the same time, the proportion belonging to the fourth moderately positive profile increased accordingly. The difference in the distribution of profiles in the population significantly differed among the three study periods (chi-sq. = 157.54; df. = 8; *p* < 0.001). There was no significant change between 2020 and 2021 (chi-sq. = 2.93; df. = 4; *p* = 0.570). The difference between the 2017 and 2020 samples is statistically significant (chi sq. = 118.44; df = 4; *p* < 0.001), as well as the difference between the 2017 and 2021 sample (chi sq. = 122.14; df = 4; *p* < 0.001).

### 3.4. Social Characteristics of the Profiles of Sexual Life Assessment

Table A1 (Appendix A below the main text) presents the characteristics of five groups representing various levels of satisfaction with sexual life based on 2021 survey data. The five groups defining sexual life evaluation profiles differed in terms of demographic and social characteristics. A number of examples can be identified where the expected frequencies in a given table cell were significantly different from those observed. For example, an overrepresentation of males in group 2 was revealed. Those living alone were more likely than expected to be in group 2, and less likely to be in the most favorable profile 5. In many other cases, relationships were consistent with predictions and involved the two extreme groups. In group 1, where sexual life ratings were worse, an overrepresentation of older people, those from poor families, those with chronic illnesses, and low rating of sexual performance was observed. Less expected results concern education and place of living. An overrepresentation of people with higher education and living in large cities appeared in group 1. Professionally active respondents rarely appeared in group 3 but often in group 5. Individuals in group 3 were also characterized by poor sexual performance ratings. In turn, non-religious people were rarely represented in group 5. In the case of sexual orientation, an overrepresentation of non-heterosexual people in group 2 was shown.

### 3.5. Barriers to Sexual Intercourse

A large fraction of respondents signaled various problems that made it difficult for them to engage in sexual intercourse. In the 2021 survey, 59.5% of respondents suggested difficulties occurred in at least one of the 16 subject areas, 35.2% more than one. Figure 2 illustrates a ranking of the most frequently reported barriers to sexual intercourse, with a comparison of two study periods conducted during the COVID-19 pandemic.

Individual difficulties accounted for between 1.2% and 31.5% of the cases. One item added to the pandemic survey questionnaire regarding the presence of others in the home (including children) appeared to be the second reported difficulty after fatigue and stress. About 5% of respondents were concerned with coronavirus infection as a potential difficulty. It is difficult to compare the frequency of reported difficulties before and during the pandemic because of changes in the questionnaire. The presence of four additional categories may have influenced the frequency of reporting other ones. However, comparing 2020 and 2021, the percentage reporting at least one of the 16 difficulties increased from 52.7% to 59.5%. Seven reasons were reported significantly more often, and only one reason was reported significantly less often. Only fear of coronavirus infection, which was rarely cited as a barrier to have sex, decreased.

The more frequently cited difficulties in 2021 compared to 2020 were: fatigue and stress (*p* = 0.012); presence of children at home (*p* = 0.031); fear that the body is not attractive (*p* = 0.044); illness of the respondent (*p* = 0.003); fear that the respondent will not perform well in sex (*p* = 0.035); partner’s lack of hygiene (*p* = 0.010); and sounds of sex from other household members or neighbors (*p* < 0.001).

Individual difficulties were also reported with different frequency by respondents from different sexual life evaluation profiles. Of the 4398 individuals eligible for analysis from the two pandemic surveys, 2453 (55.8%) reported at least one difficulty in engaging in intercourse. The frequency of reporting these concerns ranged from 52.5% in the fifth group (best rating of sexual life) to 63.2% in the second group with very poor rating of sexual life satisfaction and level of fulfilment of needs in this sphere of life. The analysis of standardized residuals indicates that only the second group should be considered as the most problematic one. In turn, in groups 1, 3, and 4, the residuals were very low and approached the threshold absolute value of 2 in group five. Those who rated their sexual life best were less likely to report difficulties in engaging in sexual intercourse.

Table 5 shows the frequency of each difficulty in the combined sample of 2020 and 2021 among those reporting at least one concern. We restricted the results to significant results. In the groups representing people with lower ratings of their sexual life, the fear of coronavirus infection, low rating of body attractiveness, partner’s abuse of alcohol, or fear of problems during intercourse were more often seen as reasons for holding back from sexual contacts, but the strongest connection seems to be associated with depression. The percentage indicating depression as an obstacle to sexual activity decreases threefold when comparing group one and five. The problems typical of the respondents who rated their sexual life better were fatigue or stress and the presence of others at home.

As part of additional analyses, the kappa was estimated. The 12 factors under study (mentioned in Table 1), year of survey, and any difficulty in engaging in the intercourse were shown to predict cluster membership poorly, indicating the need to include in further analyses other variables. The available predictors included in the threshold model especially do not provide a basis for identifying individuals in clusters 1 and 3, and the kappa was 0.220 (*p* < 0.001, however low). Ordinal regression models (PLUM) enabled us to test whether the year of the study determined profile membership after adjusting for other factors. Appendix A shows the results of the comparisons between 2017 and 2020 (Model 1) and 2020 and 2021 (Model 2). In both cases, the significant relationship with the time of data collection was confirmed. Additionally, in both cases, the significant impact of variables related to sexual life was revealed, including type of relationship (to the disadvantage of singles), frequency of intercourse (to the advantage of more frequent contact), and perception of sexual performance (to the disadvantage of its poorer rating). People in informal relationships rated their sexual life better than those who were married. The negative effect of reported difficulties hindering sexual activity was more pronounced when comparing the two pandemic surveys. Looking at both models, multivariate analysis confirmed that poorer ratings of sexual life are associated with being older, female, and chronically ill. Employment and higher material status improved ratings in the 2017–2020 period, while they had no effect on the direction of change in 2020–2021. Educational level only emerged when comparing the 2020 and 2021 surveys in favor of those with lower education. In both models, there was no association with sexual orientation, religiosity, and place of living.

## 4. Discussion

### 4.1. Interpretation of Changes

This is the first publication integrating the findings of three national cross-sectional studies, two of them conducted during the COVID-19 pandemic. To date, selected results based on 2020 survey and the first three months of the pandemic have been published [50,51].

In Poland, surveys concerning various aspects of sexuality in general population studies have been conducted for many years [48]. The exceptional circumstances resulting in social isolation during the first months of the COVID-19 pandemic created an opportunity to investigate the sexuality of Polish adults during this period and to compare the results with 2017 data, when no such restrictions existed. After the first ad hoc study had been organized in June 2020, the idea of repeating the survey after one year was born, with foresight to follow up in the final phase or after the end of the pandemic. Several papers have been published worldwide in the last two years on the impact of the pandemic on sexual relations, but they have mostly focused on one time point, usually the first months. Only a few national (and, in the light of available knowledge, also foreign) papers already published deal with cross-sectional studies repeated during the pandemic [34]. There are also a few very unique longitudinal studies [5,32]. International projects, the results of which are just beginning to be announced, have to be considered a valuable initiative [23,35]. Our analyses based on 6440 records, in total across three waves, provided an opportunity to address the pre-pandemic period while also showing trends in the months ahead. In addition, sexual satisfaction and fulfilment of sexual needs were the focus of analysis in our study. The systematic review published by Masoudi and colleagues [24] in 2022 characterized papers based mostly on lower conventional samples and evaluating changes in frequency of sexual intercourse, level of desire, number of partners, and frequency of other than partnered intercourse sexual activity (i.e., solo sex, cybersex, pornography).

In view of the various problems that appeared during social isolation, lowered mood and deterioration of the quality of life, including various aspects of sexual life, could well be expected [25]. The studies of Chatterjee and colleagues [52] showed that at the time of social isolation caused by the first months of the pandemic, 32.9% of respondents experienced an increased level of stress, 34.9% reported depression, and 39.5% had anxiety symptoms. From the first weeks of the pandemic, publications began to appear indicating the need to protect oneself against coronavirus infection, as well as the psychological consequences of living in isolation [53]. A higher level of overall anxiety may have resulted in lower sexual satisfaction or caused sexual disorders, which may additionally potentiate mental symptoms, finally leading to reduced interest in sex [54]. The analyses we conducted did not embrace an in-depth assessment of mental health. However, depression, as a disturbing factor in taking up sexual contacts, was reported by 25.2% of respondents, having pronounced problems with their sexual lives (the so-called least favorable profile). Comparison of responses to the key questions in the three study periods supported an inference of a systematic deterioration in satisfaction with sexual life and the level of fulfilment of needs in this sphere of life, as well as increasingly poorer assessment of one’s sexual performance. Only the frequency of sexual intercourse periodically increased at the beginning of the pandemic, but in 2021 it was already below the 2017 level. The initial increase may be related to the fact that partners had an opportunity to spend more time together, and participate in common activities. Moreover, it cannot be ruled out that increased interest in sex was associated with acting out stressful situations and the manifestation of symptoms of the compulsive sexual behavior disorder [55,56]. A similar study of increased interest in sex in the context of mental conditions was conducted in Italy [29], where health and behavior before the COVID-19 pandemic and during the quarantine were compared. Although worsened well-being was recorded among the studied persons (71%), the level of sexual desire was not diminished. In our study, the level of sexual need proved to be the most stable indicator when comparing the three study periods.

The factors which disturbed sexual activity mentioned by the study participants surveyed during the pandemic were mainly fatigue and stress, presence of other persons (including children) at home and lack of privacy, as well as uncertainty as to one’s physical attractiveness and the respondent’s illness. A Swiss study has shown that people who strongly agreed that their housing lacked comfort reported great increases in psychological strain [57]. A new problem emerged during the pandemic, namely fear of COVID-19 infection, as highlighted in other papers [58]. In the study of Ko and colleagues [30], it was found that fear of COVID-19 infection was often connected with limiting sexual activity, which was perceived as a potential source of infection, both in random sexual contacts and in permanent relationships. However, new studies may indicate a decreasing concern about infection risk, as confirmed by our analysis.

### 4.2. Sexual Life Assessment Profiles

One advantage of this study was its inventive methodological approach. An attempt was made to identify groups of persons who presented similar opinions about various aspects of sexuality. Five groups (clusters) of respondents were identified, characterised by distinct sexual life assessment profiles created on the basis of four questions, and externally validated. In the process of identifying those profiles, it was demonstrated that nearly half of the respondents gave a very good assessment of their sexual life in all three study periods in the four analysed aspects. However, in 2021 for more than one in three respondents, this assessment was not quite so positive. Some studies have used composite indices to assess the quality of sexual life, but these referred to clinical, rather than subjective, assessment [31,59].

Using the cluster analysis, an interesting group was detected (group 2) of persons who had very high sexual needs, but were unable to fulfil them, being generally dissatisfied with their sexual life. Moreover, the occurrence of this group increased in 2017–2021, which could be a result of living conditions and the difficulties experienced during the pandemic. This group consisted mainly of men aged 30–49, living single, with lower than secondary education, and working. On the basis of combined 2020 and 2021 samples, they were more likely than others to report fear of unsatisfactory performance at sex. Acute stress, which appeared during the period of social isolation, could give rise to men’s concerns about sexual performance. Stress, as the underlying cause for erectile disorders, as well as premature ejaculation, are well documented [60,61], also during the COVID-19 pandemic [62,63]. In addition, unfavourable conditions for creating an intimate atmosphere could lead to reducing the frequency of sexual intercourse attempts, giving rise to a feeling of unfulfillment and general dissatisfaction with sexual life, especially in men with a high level of sexual needs.

On the other hand, the occurrence of group 4, that is, persons having a rather good (but not the best) assessment of all four aspects of sexual life, increased. Married women prevailed in this group, while the presence of children and other persons at home and lack of intimacy was reported as a barrier to sexual intercourse. Other studies documented a greater deterioration of sexual life among working women, compared to non-working women [38]. Fear of unemployment and the inability to meet living expenses caused increased levels of stress, which affected both genders and led to decreased quality of sexual life [64,65,66]. Poor body image was quoted as the third major barrier to sexual intercourse during the pandemic. This may be a result of paying less attention to look attractive during a prolonged stay at home. Many factors may have influenced the change in the perception of body attractiveness including pandemic-related stress, changing health habits, worse material conditions, limited access to cosmetic services, and damaging social media messages [67,68].

The studies we described emphasised the socio-demographic structure of our sample, thereby also drawing attention to the problems of persons not living in a permanent relationship (20.9% of respondents aged 18–70 years in 2021). These persons were strongly represented in group 1 and 2, in which the assessment of sexual life was the worst. The potentially negative effect of loneliness on mental health during the pandemic was also discussed in other studies conducted among the Polish population, which did not, however, include sexual life [69].

### 4.3. Limitations and Implications of This Study

The conducted analyses have a number of shortcomings, but we hope that their strengths will outweigh them. In a comparison of the 2017 to 2020–2021 studies, it is difficult to determine what was caused by life during isolation and what was the result of a natural, gradual process of habitual and behavioral changes. The survey procedure from 2020 and 2021 was identical (CAWI), and the received samples did not differ in terms of basic characteristics. However, when concluding about the differences between the time before and during the pandemic, it is necessary to keep in mind the changing method of data collection and the characteristics of the respondents. Moreover, repeated cross-sectional studies do not track individual changes (as longitudinal studies do). However, they are widely used for trend analyses of health indicators.

The cluster analysis as an exploratory method does not guarantee the only right solution. However, while characterizing the emerged groups, we pointed out the directions of further analyses in order to obtain an even better classification and set of factors predicting group membership. Having the extended set of predictors, it would be worthwhile to apply advanced statistical models, such as threshold models. A significant number of the factors identified in the introduction under the operational definition of sexual life have been analyzed, but not all. In the future, it would be advisable to take into account a larger number of factors related to sexual life (also clinical indicators), which would help explain “atypical” profiles, such as group 2.

Nevertheless, the present study is the first to examine sexual life of the Polish population during the COVID-19 pandemic. We report that the quality of sexual life worsened with the duration of the pandemic. Therefore, further investigation of strategies to improve both sexual and mental health during the pandemic should be considered. The obtained results may guide further research in Poland and other countries, but also support those actively involved in counseling, including medical doctors, sexologists, and psychologists.

## 5. Conclusions

Ad hoc surveys conducted during the COVID-19 pandemic do not provide a complete picture of changes in attitudes toward sexual life. Some indicators may have changed periodically in the first months and stabilized in subsequent months of the pandemic. Two studies repeated in Poland in 2020–2021 in a group of people aged 18–70 years indicate overall deterioration in the evaluation of their sexual lives. There has been a marked decline in the proportion of the population rating their sexual life very well, coexisting with increase of moderately well rating. An increasing number of people reported the existence of factors that make it difficult to have sexual intercourse. The findings provide a basis for further tracking of changes in sexual lives in the context of the pandemic experiences.

## Figures and Tables

**Figure 1 ijerph-19-04110-f001:**
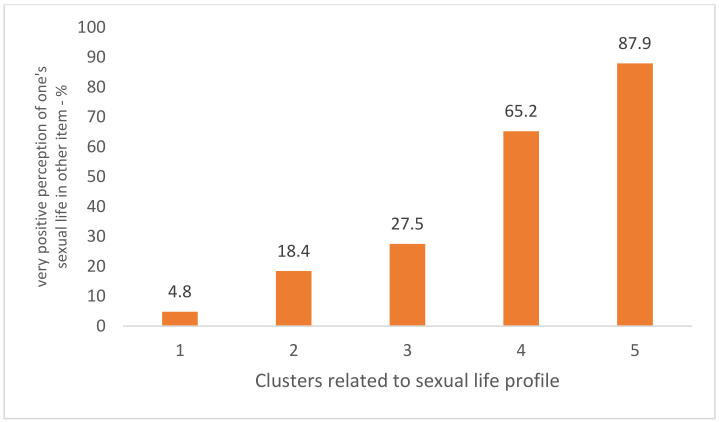
Sexual life assessment profiles suggested by cluster analysis ranked by the percentage of respondents who rated this part of life very well in the independent item (2021 survey).

**Figure 2 ijerph-19-04110-f002:**
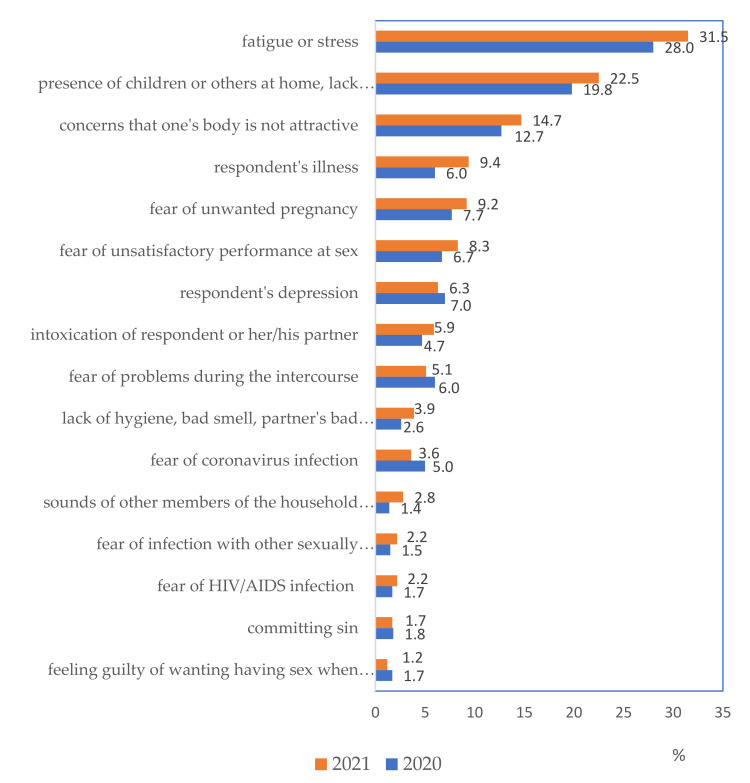
Problems causing difficulties in sexual life during the pandemic (% of those having at least one problem).

**Table 1 ijerph-19-04110-t001:** Characteristics of respondents participating in the 2017, 2020, and 2021 surveys.

	1—2017		2—2020		3—2021		*p*		
	*n*	%	*n*	%	*n*	%	1, 2, 3	1, 2	2, 3
Total	2042	100	2418	100	1980	100			
Gender									
Male	1016	49.8	1223	50.6	990	50.0	0.851	0.584	0.702
Female	1026	50.2	1195	49.4	990	50.0			
Age (years)									
18–29	558	27.3	445	18.4	348	17.6	<0.001	<0.001	0.774
30–49	928	45.5	1078	44.6	894	45.1			
50–70	556	27.2	895	37.0	738	37.3			
Level of education									
Lower than secondary	752	36.8	1109	45.8	880	44.4	<0.001	<0.001	0.628
Secondary	687	33.7	555	23.0	471	23.8			
Higher	603	29.5	754	31.2	629	31.8			
Place of living									
Rural areas	767	37.6	1005	41.6	770	38.9	0.022	0.006	0.198
Smaller towns	1046	51.2	1024	46.4	961	48.5			
Large cities	229	11.2	289	12.0	249	12.6			
Employment status									
Working	1499	75.5	1373	58.3	1165	60.2	<0.001	<0.001	0.188
Other	487	24.5	984	41.7	769	39.8			
Family material status									
Low	200	10.4	554	23.6	371	19.2	<0.001	<0.001	0.002
Average	1094	56.7	1097	46.6	959	49.6			
High	635	32.9	700	29.8	604	31.2			
Religious beliefs									
Believers	1756	91.7	1915	85.8	1511	83.2	<0.001	<0.001	0.025
Disbelievers	159	8.3	318	14.2	305	16.8			
Status of relationship									
Single	513	25.1	457	18.9	413	20.9	<0.001	<0.001	0.132
Married	1041	51.1	1265	52.5	1040	52.7			
Informal	486	23.8	691	28.6	522	26.4			
Sexual orientation									
Heterosexual	1926	94.3	2243	92.8	1830	92.4	0.037	0.036	0.670
Non-heterosexual	116	5.7	175	7.2	150	7.6			
Self-rated sexual performance									
Poor	93	4.7	215	8.9	213	10.9	<0.001	<0.001	<0.001
Average	316	16.0	505	20.9	445	22.8			
Good	885	44.9	1007	41.8	840	43.0			
Very good	678	34.4	684	28.4	455	23.3			
Last sexual intercourse									
This week	976	47.7	1265	52.3	783	39.5	<0.001	<0.001	<0.001
This month	249	12.2	315	13.0	224	11.3			
This year	95	4.7	188	7.8	152	7.7			
Earlier or missing data	722	35.4	650	26.9	821	41.5			
Chronic conditions									
No	1578	78.0	1519	63.5	1236	63.2	<0.001	<0.002	0.831
Yes	446	22.0	873	36.5	720	36.8			

*p* for the chi-squared test.

**Table 2 ijerph-19-04110-t002:** Evaluation of own sexual life in the three study periods (% of positive and negative responses).

Item		Year of Data Collection			*p*			
		1—2017	2—2020	3—2021	1, 2, 3	1, 2	1, 3	2, 3
What is the role of sex in your life at present?	−	19.0	15.0	18.2	<0.001	<0.001	0.841	0.001
	+	54.4	60.3	55.1				
How satisfied have you been with your sexual life over the past 2–3 months?	−	9.7	15.9	18.3	<0.001	<0.001	<0.001	0.094
	+	66.1	62.3	61.1				
How do you assess the level of your sexual needs over the past 2–3 months?	−	17.1	16.6	16.5	0.246	0.340	0.070	0.630
	+	30.7	32.7	34.0				
Have your sexual needs been fulfilled over the past 2–3 months?	−	13.3	16.6	18.9	0.002	0.009	<0.001	0.021
	+	69.6	67.0	67.2				

− two most negative answers; + two most positive answers; *p* for chi-sq. test.

**Table 3 ijerph-19-04110-t003:** Evaluation of sexual life in 2021 according to sexual life profiles (% of positive and negative responses).

Item		Sexual Life Assessment Profile					*p*
		1—Worst	2	3	4	5—Best	
What is the role of sex in your life at present?	−	87.2	2.8	75.1	4.1	0.0	<0.001
	+	0.0	57.1	0.9	45.5	97.8	
How satisfied have you been with your sexual life over the past 2–3 months?	−	74.4	58.0	16.9	0.2	0.2	<0.001
	+	1.1	3.7	23.1	84.0	98.6	
How do you assess the level of your sexual needs over the past 2–3 months?	−	51.7	1.2	77.3	9.2	0.0	<0.001
	+	11.7	53.7	0.4	8.2	66.9	
Have your sexual needs been fulfilled over the past 2–3 months?	−	95.6	57.7	6.7	0.0	0.2	<0.001
	+	0.0	4.9	52.0	93.1	98.1	

− two most negative answers; + two most positive answers; *p* for chi-sq. test

**Table 4 ijerph-19-04110-t004:** Prevalence of sexual life assessment profiles in three samples.

Sexual Life Assessment Profile	2017			2020			2021		
	*n*	%	Res.	*n*	%	Res.	*n*	%	Res.
1—worst	167	8.18	−0.39	196	8.11	−0.55	180	9.09	1.01
2	292	14.30	−1.34	380	15.72	0.31	324	16.36	1.02
3	244	11.95	0.78	263	10.88	−0.71	225	11.36	0.00
4	377	18.46	−7.44	752	31.10	3.86	611	30.86	3.29
5—best	962	47.11	6.91	827	34.20	−2.81	640	32.32	−3.91

Res.—standardized residual.

**Table 5 ijerph-19-04110-t005:** Selected difficulties in initiating sexual intercourse (%) in groups representing different profiles of sexual life assessment (combined 2020 and 2021 data; *n* = 2453 respondents reporting at least one difficulty).

Difficulties in Sexual Intercourse	Sexual Life Assessment Profile					*p*
	1—Worst	2	3	4	5—Best	
Fear of HIV/AIDS infection	4.3	5.8	3.8	1.9	3.2	0.009
Fear of coronavirus infection	13.3	11.5	6.2	6.0	6.8	<0.001
Lack of hygiene, bad smell, partner’s bad breath	12.4	7.0	6.2	4.7	3.9	<0.001
Fear of problems during the intercourse	16.7	15.1	11.4	7.6	6.9	<0.001
Intoxication of respondent or her/his partner	11.9	12.6	6.6	10.1	7.3	0.007
Respondent’s depression	25.2	17.5	14.9	8.1	7.8	<0.001
Fear of unsatisfactory performance at sex	17.1	20.4	11.4	11.4	10.6	<0.001
Respondent’s illness	18.6	12.4	19.7	12.7	14.2	0.012
Concerns that one’s body is not attractive	35.7	27.0	26.6	22.5	20.8	<0.001
Presence of children or others at home, lack of intimacy	21.9	29.9	27.0	41.8	46.5	<0.001
Fatigue or stress	42.9	49.9	54.7	58.3	51.8	0.001

*p* for chi-sq test.

## Data Availability

The data are owned by Warsaw University and are not to be made freely publicly available.

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
