# Peer review of "The Effect of the COVID-19 Pandemic on the Assessment of Sexual Life—Repeated Cross-Sectional Surveys among Polish Adults in 2017, 2020 and 2021"

_ijerph, 2022, doi:10.3390/ijerph19074110_

Round 1

Reviewer 1 Report

The article is interesting, even if it does not appear as particularly original.

However, it is nice to read these results in a particular East European population.

The main problem of the manuscript is a weak capacity of considering previous literature on the same topic of covid / lockdown / longcovid, etc. For this reason, the article must be carefully rewritten confronting these results with those previously published:

doi: 10.1016/j.jsxm.2020.10.008. 

Author Response

The article is interesting, even if it does not appear as particularly original.

However, it is nice to read these results in a particular East European population.

The main problem of the manuscript is a weak capacity of considering previous literature on the same topic of covid / lockdown / longcovid, etc. For this reason, the article must be carefully rewritten confronting these results with those previously published:

THANK YOU VERY MUCH FOR FINDING OUR ARTICLE INTERESTING.  WE HAVE INCREASED THE NUMBER OF REFERENCES TO PAPERS BY OTHER AUTHORS. IN THE INTRODUCTION, WE POINTED OUT THE VARIETY OF TOPICS LINKING HUMAN SEXUALITY IN THE CONTEXT OF COVID-19 PANDEMIC. WE HAVE ALSO INCLUDED MENTIONED IN THE REVIEW.

Reviewer 2 Report

This is an interesting article, but its limitations make it a poor contribution to the journal. I present my arguments:

  1. Please rewrite Abstract: Please avoid ambiguous language “some aspects”, and replace this with objective variables. Include clear objectives. How did Results confirmed a potential negative 18 impact of the COVID-19 pandemic? Include conclusion.
  2. Introduction: authors should reduce the information of Quality of Life (until line 85) and focus more on sexual life.
  3. Authors should provide an operational definition of sexual life.
  4. Authors must provide more research results on sexual variables and covid-19. There are many results, including from Poland, that have not been cited here.
  5. Please rewrite objective clearly. Avoid from subjective terms such as “attempt to”.
  6. I believe the questions posed to not constitute the construct of sexual life, rather, some self-assessed impression of sexual aspects that cannot be seen as a construct, even if multidimensional, therefore, reliability analysis is redundant.
  7. Please refrain from presenting the information in points.
  8. Please include an implications section.
  9. Data analysis and results are not feasible, lack validity, and descriptive and inferential analyses are insufficient. More robust analyses should be performed, such as moderation or mediation analyses.

The main problem: It is not possible to demonstrate that changes occurred are directly attributable to the COVID-19 pandemic, since this was not controlled.

Author Response

This is an interesting article, but its limitations make it a poor contribution to the journal.

THE ARTICLE WAS WRITTEN FOR THE IJERPH SPECIAL ISSUE ON SEXUAL LIFE IN A PANDEMIC AND SHOWS THE PERSPECTIVE OF POLAND AS A COUNTRY WITH EXCEPTIONALLY UNFAVOURABLE CONTAMINATIONS AND DEATH STATISTICS.

I present my arguments:

  1. Please rewrite Abstract: Please avoid ambiguous language “some aspects”, and replace this with objective variables. Include clear objectives. How did Results confirmed a potential negative impact of the COVID-19 pandemic? Include conclusion.

THE PHRASE "SOME ASPECTS" WAS REMOVED BOTH FROM THE ABSTRACT AND AT THE END OF THE INTRODUCTION WITH A MORE DETAILED DESCRIPTION. THE AIM OF THE STUDY WAS SPECIFIED, BUT ONLY IN THE MAIN TEXT DUE TO THE WORD LIMIT IN THE ABSTRACT.  IT WAS EXPLAINED THAT THE NEGATIVE EFFECT OF THE PANDEMIC IS EXPRESSED BY A LOWER PERCENTAGE OF PERSONS CLASSIFIED INTO THE PROFILE (CLUSTER) OF THE BEST ASSESSMENT OF SEXUAL LIFE. CONCLUSION REWORDED

  1. Introduction: authors should reduce the information of Quality of Life (until line 85) and focus more on sexual life.

THE INTRODUCTION INCLUDES NOW MORE REFERENCES TO RESEARCH RESULTS FROM DIFFERENT COUNTRIES AND THE PARAGRAPH ON QUALITY OF LIFE HAS BEEN REDUCED.

  1. Authors should provide an operational definition of sexual life.

THIS DEFINITION IS NOW PROVIDED IN THE INTRODUCTION (LINES 60+). IT GAVE A BASIS FOR QUOTING PAPERS RELATING TO THE PANDEMIC (AS SUGGESTED ABOVE), WHERE VARIOUS ASPECTS OF SEX LIFE WERE EXAMINED.

  1. Authors must provide more research results on sexual variables and covid-19. There are many results, including from Poland, that have not been cited here.

IN THE PREVIOUS VERSION, SEVERAL PAPERS FROM POLAND WERE CITED, BUT NEW ONES WERE ADDED. FIRST OF ALL WE SEARCHED FOR PAPERS FROM DIFFERENT COUNTRIES WHERE STUDIES ON SEXUALITY WERE REPEATED DURING THE PANDEMIC (LIKE OURS), WE HAD TO SKIP MANY SINGLE (AD HOC) STUDIES BASED ON DATA FROM THE BEGINNING OF THE PANDEMIC.

             Please rewrite objective clearly. Avoid from subjective terms such as “attempt to”.

TO BETTER CLARIFY THE OBJECTIVES, WE ADDED RESEARCH QUESTIONS.

NO TERM SUCH AS "ATEMPT TO" IN THIS VERSION

  1. I believe the questions posed to not constitute the construct of sexual life, rather, some self-assessed impression of sexual aspects that cannot be seen as a construct, even if multidimensional, therefore, reliability analysis is redundant.

THAT IS A VERY GOOD POINT. WE HAVE EVEN ADDED THE WORD "ASSESSMENT" IN THE TITLE OF THE PAPER. WE HAVE DELETED COMMENTS ON THE RELIABILITY OF THE SCALE, WHICH COULD HAVE BEEN ANALYSED ALTERNATIVELY. THE MEANS INDEX IS NOT PRESENTED AS WELL. THE ADVANTAGE OF OUR STUDY (AND THE ADVANTAGE OVER A SUMMARY INDEX OF 4 QUESTIONS) IS TO SHOW THAT PART OF THE POPULATION EVALUATES ONE ASPECT POSITIVELY, WHILE OTHER LESS POSITIVELY.

  1. Please refrain from presenting the information in points.

IN SOME PARTS, THE POINTS MAKE THE ARTICLE EASIER TO READ (E.G. SCOPE OT THE QUESTIONNAIRE ) AND SHORTEN THE DESCRIPTION. THE BULLET POINT HAS BEEN REPLACED BY CONTINUOUS TEXT IN 3.5 (BARRIERS).

  1. Please include an implications section.

THE IMPLICATIONS ARE DESCRIBED IN 4.3 TOGETHER WITH THE LIMITATIONS. THIS SECTION HAS BEEN ENLARGED, BECAUSE PREVIOUSLY WE FOCUSED ON THE IMPLICATIONS FOR FURTHER RESEARCH.

  1. Data analysis and results are not feasible, lack validity, and descriptive and inferential analyses are insufficient. More robust analyses should be performed, such as moderation or mediation analyses.

STATISTICAL METHODS DEPEND ON THE OBJECTIVES OF THE STUDY AND THE TYPE OF OUTCOME VARIABLE. IN OUR CASE, THE MAIN OUTCOME VARIABLE IS CLUSTER (PROFILE) MEMBERSHIP. AN ORDINAL VARIABLE THAT DOES NOT EVEN HAVE THE ATTRIBUTES OF A QUASI-CONTINUOUS VARIABLE. IT IS THEN DIFFICULT TO APPLY CLASSICAL REGRESSION METHODS (AS FOR CONTINUOUS VARIABLES). WHEN PREPARING THIS ARTICLE, WE ESTIMATED THE MULTIVARIATE ORDINAL REGRESSION MODEL SO CALLES PLUM REGRESSION. ONLY KAPPA COEFFICIENT HAS BE PRESENTED SO FAR. MORE RESULTS ARE NOW INCLUDED WITH DETAILED MODEL THE MULTIVARIATE MODEL CONFIRMED  THE SIGNIFICANT EFFECT OF "YEAR OF STUDY" AFTER ADJUSTING FOR OTHER FACTORS AND COVARIATES. TWO PLUM REGRESSION MODELS ARE NOW INCLUDED AS ADDITIONAL ELECTORAL MATERIAL (TABLE S2). AT THE END OF THE RESULTS SECTION THESE MODELS ARE DESCRIBED.  

The main problem: It is not possible to demonstrate that changes occurred are directly attributable to the COVID-19 pandemic, since this was not controlled.

WE WERE AWARE OF THIS, AND SO THE SECTION ON LIMITATIONS BEGINS WITH A SIMILAR REMARK. THE PAPER PRESENTS A COMPARISON OF THREE TIME POINTS, TWO OF THEM DURING THE PANDEMIC. AND ONLY IN THIS CONTEXT ONE CAN CONCLUDE ON THE DIFFERENCES. ANALYSES WERE ADJUSTED FOR A DOZEN RESPONDENT CHARACTERISTICS. THE ESTIMATION OF THE PLUM REGRESSION MODELS HAS BEEN ADDED.A NUMBER OF VARIABLES THAT WERE DIRECTLY RELATED TO THE PANDEMIC WERE INCLUDED. FOR EXAMPLE, IT WAS SHOWN TO WHAT EXTENT THE PERMANENT PRESENCE OF OTHERS AT HOME MADE SEXUAL ACTIVITY MORE DIFFICULT. WHEN ASKING THE QUESTIONS THE TIME FRAME WAS ALSO CONTROLLED (EG. LAST 2-3 MONTHS IN JUNE 2020 SURVEY). 

Reviewer 3 Report

The manuscript entitled “Effect of COVID-19 Pandemic on Sexual Life – Repeated Cross-Sectional Surveys Conducted among Adult Poles in 2017, 2020 and 2021” is an intriguing study that attempts to examine the effect of COVID-19 pandemic on sexual life of adult poles. The authors use rich data set from three rounds of survey conducted in 2017, 2020, and 2021, and employ different methods to categorize the respondents into several groups according to their evaluations of sexual life. The authors conclude that there is a potential negative effect of COVID-19 on respondents’ sexual life. The article is mostly well-written, and structured. The literature review is sufficient, and the results are interesting. However, there are some issues that need to be resolved before the paper can be processed further. My comments and suggestions are as follows:

One main concern with the survey data is that the survey respondents are not the same in the three time periods. Furthermore, there are some significant socioeconomic differences in the respondents in the three surveys. On the other hand, when we look at the results, the differences are not very large (although statistically significant). It is possible that the changes being shown here are due to heterogeneity of respondents (shown in section 3.1) in the various surveys (the facts that respondents were different each time). In the absence of a regression analysis (such as SEM) to control for these differences, the authors cannot be certain of the causality i.e., the observed changes are really caused by the pandemic, and not the other factors. I would suggest the authors to include a sounds regression analysis to control for other factors to isolate the effects of pandemic.

What is the external and internal validity of scales used to measure “approach to sex life”. The authors cited a previous study which does not seem to be a peer-reviewed study, and its published in Polish language (hence difficult for me to evaluate). Please explain.

Please add a Table of descriptive statistics of respondents for the three survey rounds.

Figure 1: The figure is not self-explanatory. What these different clusters are, and what differences are being shown here? I suggest moving this Figure after Table 3.

Minor correction:

Line 141. “How satisfied are you with our sex life over the past 2–3 months / 12 months? with five categories of answers, from completely dissatisfied to very satisfied.” Here is should be “your” instead of “our”.

Author Response

The manuscript entitled “Effect of COVID-19 Pandemic on Sexual Life – Repeated Cross-Sectional Surveys Conducted among Adult Poles in 2017, 2020 and 2021” is an intriguing study that attempts to examine the effect of COVID-19 pandemic on sexual life of adult poles. The authors use rich data set from three rounds of survey conducted in 2017, 2020, and 2021, and employ different methods to categorize the respondents into several groups according to their evaluations of sexual life. The authors conclude that there is a potential negative effect of COVID-19 on respondents’ sexual life. The article is mostly well-written, and structured. The literature review is sufficient, and the results are interesting. However, there are some issues that need to be resolved before the paper can be processed further. My comments and suggestions are as follows:

THANK YOU VERY MUCH FOR THE OVERALL POSITIVE ASSESSMENT OF OUR PAPER AND FOR APPRECIATING THE SIZE OF THE SAMPLE AS WELL AS THE REFERENCES TO LITERATURE.  

One main concern with the survey data is that the survey respondents are not the same in the three time periods. Furthermore, there are some significant socioeconomic differences in the respondents in the three surveys. On the other hand, when we look at the results, the differences are not very large (although statistically significant). It is possible that the changes being shown here are due to heterogeneity of respondents (shown in section 3.1) in the various surveys (the facts that respondents were different each time). In the absence of a regression analysis (such as SEM) to control for these differences, the authors cannot be certain of the causality i.e., the observed changes are really caused by the pandemic, and not the other factors. I would suggest the authors to include a sounds regression analysis to control for other factors to isolate the effects of pandemic.

UNFORTUNATELY, WE DO NOT HAVE THE RESULTS OF LONGITUDINAL STUDIES COMBINING THE PERIOD BEFORE AND DURING THE PANDEMIC BECAUSE THE PANDEMIC WAS NOT EXPECTED. SUCH COMPARISONS OF INDEPENDENT CROSS-SECTIONAL STUDIES ARE OFTEN PUBLISHED. ONE ADVANTAGE OF OUR STUDY IS THE ORGANISATION OF THREE ROUNDS BY THE SAME TEAM AND THE USE OF THE SAME QUESTIONNAIRE, WHICH INCREASES COMPARABILITY OF RESULTS. AS SUGGESTED IN THIS AND THE OTHER REVIEW, WE HAVE INCLUDED ANALYSES ADJUSTED FOR ALL VARIABLES UNDER STUDY. THEY SHOW THAT THE "YEAR OF SURVEY" EFFECT IS SIGNIFICANT IF WE ARE LOOKING FOR THE INDEPENDENT PREDICTOR OF  CLUSTER (PROFILE) MEMEBRSHIP (MEANS ADJUSTED FOR DIFFERENCES IN RESPONDENTS' CHARACTERISTICS). ORDINAL REGRESSION WAS USED AS A KIND OF THRESHOLD MODEL. TRADITIONAL LINEAR OR LOGISTIC REGRESSION AS WELL AS SEM (PATH) MODELS ARE NOT APPLICABLE FOR SUCH AN OUTCOME. THE MAIN OBJECTIVE IS TO ASSESS THE FREQUENCY OF POFILES (CLUSTERS) IN THE POPULATION, WHICH WOULD BE DIFFICULT TO INVESTIGATE WITH A PATH/SEM MODEL.

What is the external and internal validity of scales used to measure “approach to sex life”. The authors cited a previous study which does not seem to be a peer-reviewed study, and its published in Polish language (hence difficult for me to evaluate). Please explain.

THE FOUR MAIN QUESTIONS USED I OUR PAPER HAVE BEEN APPLIED IN THE RESEARCH ON SEXUALITY OF POLES SINCE 1997. THEY ARE RELATED TO THE GENERAL EVALUATION (ROLE OF SEX IN LIFE, GENERAL SATISFACTION) AND SEXUAL NEEDS (LEVEL, DEGREE OF FULFILMENT). IN SOME PREVIOUS ANALYSES, WE TREATED THESE QUESTIONS AS AN INDEX, SINCE ITS PSYCHOMETRIC PROPERTIES ARE REALLY GOOD. AS POINTED OUT IN ANOTHER REVIEW, THE USE OF SUCH AN INDEX TOGETHER WITH PROFILES MAY BE A SOURCE OF CONFUSION. WE HAVE DELETED COMMENTS ABOUT THE RELIABILITY OF THE SCALE AND ITS USE IN OTHER FORMER PUBLICATIONS. THE ADDED VALUE OF PRESENT STUDY IS TO GROUP PEOPLE WHO ARE SIMILAR, AND TO DETECT „NON-STANDARD” RESPONDENTS. THE TRADITIONAL SUMMARY INDEXES ASSUME COHERENCE OF ANSWERS TO ALL 4 QUESTIONS (ALWAYS GOOD, AVERAGE OR POOR). WE SHOWED THAT SOME PEOPLE FEEL THAT SEX IS IMPORTANT FOR THEM, BUT CANNOT SATISFY THEIR NEEDS OR FEEL SATISFIED AT THE LOW LEVEL OF NEEDS. THE PROPORTION OF SUCH GROUPS INCREASED DURING THE PANDEMIC PERIOD AND IT SEEMS TO BE AN IMPORTANT RESULT.

THE VALIDATION OF THE CLUSTERS THEMSELVES RESULTS FROM THE METHOD AND THE POST HOC ANALYSIS. THE PROFILES ARE DISTINC, THEY DIFFER FOR EACH OF THE 4 QUESTIONS.  

please add a Table of descriptive statistics of respondents for the three survey rounds.

SUCH A TABLE WAS IN THE APPENDIX BEFORE AS A1 BELOW THE MAN TEXT AND BEFORE REFERENCES. IT HAS BEEN MOVED TO THE MAIN TEXT (NOW TABLE 1).

Figure 1: The figure is not self-explanatory. What these different clusters are, and what differences are being shown here? I suggest moving this Figure after Table 3.

THE FIGURE WAS INTENDED TO HELP RANKING THE CLUSTERS FROM THE WORST TO THE BEST RATING OF SEXUAL LIFE. CLUSTER ANALYSIS PROVIDE RESULTS IN RANDOM ORDER. THIS WAS DONE BY MEANS OF A QUESTION PLACED IN ANOTHER PART OF THE QUESTIONNAIRE, AS DESCRIBED IN THE METHODS. THE TITLE OF THE FIGURE AND TITLE ON THE AXIS HAVE BEEN CHANGED. IT IS NOT JUSTIFIED TO MOVE THIS FIGURE FORWARD, AS THE PRESENTATION OF FURTHER RESULTS DEPENDS ON THE RANKING OF THE CLUSTERS.

Minor correction:

Line 141. “How satisfied are you with our sex life over the past 2–3 months / 12 months? with five categories of answers, from completely dissatisfied to very satisfied.” Here is should be “your” instead of “our”.

THANK YOU FOR NOTICING THIS TYPO. HAS BEEN CORRECTED.

Round 2

Reviewer 1 Report

the ms appears improved

Author Response

We would like to thank you very much for accepting the changes made to the manuscript.  The manuscript was proofread once again.

Reviewer 2 Report

Thank you for implementing all the requested changes to the article. They have improved the overall quality of the manuscript, but the main problem was not corrected: It is not possible to demonstrate that changes occurred are directly attributable to the COVID-19 pandemic, since this was not controlled. Therefore, assessing the effect of the COVID-19 on sexual life was not achieved. 

Author Response

We fully understand this concern and have therefore addressed it in the discussion. Many studies now refer to the impact of the pandemic because they were conducted during it. Our 2020 and 2021 surveys was aimed at the general population and did not look at the possible impact of infection. The manuscript was proofread again and language errors were removed.